# Transcription Profile of Auxin Related Genes during Positively Gravitropic Hypocotyl Curvature of *Brassica rapa*

**DOI:** 10.3390/plants11091191

**Published:** 2022-04-28

**Authors:** Chitra Ajala, Karl H. Hasenstein

**Affiliations:** 1Biology Department, University of Louisiana at Lafayette, Lafayette, LA 70504, USA; chitrapandita@gmail.com; 2Cemvita Factory, 9350 Kirby Drive, Suite 100, Houston, TX 77054, USA

**Keywords:** *Brassica rapa*, Solid Phase Gene Extraction (SPGE), germination, gravitropism, hypocotyl, root

## Abstract

Unlike typical negative gravitropic curvature, young hypocotyls of *Brassica rapa* and other dicots exhibit positive gravitropism. This positive curvature occurs at the base of the hypocotyl and is followed by the typical negative gravity-induced curvature. We investigated the role of auxin in both positive and negative hypocotyl curvature by examining the transcription of *PIN1*, *PIN3*, *IAA5* and *ARG1* in curving tissue. We compared tissue extraction of the convex and concave flank with Solid Phase Gene Extraction (SPGE). Based on Ubiquitin1 (*UBQ1*) as a reference gene, the log (2) fold change of all examined genes was determined. Transcription of the examined genes varied during the graviresponse suggesting that these genes affect differential elongation. The transcription of all genes was upregulated in the lower flank and downregulated in the upper flank during the initial downward curving period. After 48 h, the transcription profile reversed, suggesting that the ensuing negative gravicurvature is controlled by the same genes as the positive gravicurvature. High-spatial resolution profiling using SPGE revealed that the transcription profile of the examined genes was spatially distinct within the curving tissue. The comparison of the hypocotyl transcription profile with the root tip indicated that the tip tissue is a suitable reference for curving hypocotyls and that root and hypocotyl curvature are controlled by the same physiological processes.

## 1. Introduction

Gravitropism is defined as a directional growth response of plants to gravity. This force affects the redistribution of statoliths (starch-filled amyloplasts) in sensory cells and determines the positioning of the plant axis during and after germination. Seedling roots grow towards gravity (aka positive gravitropism) and stems exhibit negative gravitropism and grow in the opposite direction. Although hypocotyls eventually respond as negatively gravitropic, we have previously shown that young hypocotyls of *Brassica rapa* and other dicotyledonous plants exhibit positive gravitropism. This reorientation is independent of the root cap and amplifies the root graviresponse [1]. The mechanism of this atypical, early curvature has not yet been studied. Based on the hypothesis that this curvature is auxin dependent, we investigated the transcription profile of auxin-responsive genes in the curving hypocotyl tissue.

The phytohormone indole-3-acetic acid (IAA) is the main growth regulator in plants that affects differential elongation [2] and is the main agent in complex processes that includes transport [3], and interaction with other phytohormones [4,5,6]. Auxin effects depend on metabolic activity [7], cytoskeletal interaction [8,9], ions [10,11,12,13], and light [14,15,16,17,18]. Auxin regulates primary and lateral root growth [19,20], branching [21], and differentiation [22]. The coordinated response to the integrative effect of these factors determines the rate and degree of curvature. Therefore, differential growth responses depend on auxin-mediated processes and gene expression.

Auxin is synthesized predominantly in meristematic tissue such as shoot and root apices [23], developing seeds [24] and young leaves [25] and transported away from the site of synthesis. Auxin originating from the shoot apex is basipetally (rootward) transported to the stem tissue and basal leaves. In roots, auxin moves through the stele towards the root apex and is redistributed such that basipetal, shootward, transport takes place from the root tip through cortical tissue. The bi-directional auxin transport in roots forms the basis of the “reverse fountain model” [2]. The gravisensitive re-distribution of auxin in roots is initiated by amyloplast relocation and depends on PIN proteins. These proteins determine the direction of IAA flow based on their localization in the plasma membrane [26]. PIN proteins represent a family of proteins that are expressed in specific tissues [27].

PIN1 is typically present in xylem associated cells in shoots and roots [23] and is essential for efflux of the auxin anion from vasculature to root apices via protophloem cells [28,29]. In roots, PIN3, 4, and 7 are localized on the plasma membrane of different tiers of columella cells [30]; they direct the auxin flow from columella cells to the root cortex. PIN3 and PIN7 move to the plasma membrane along the lower side of columella cells in horizontally reoriented roots [31], which results in asymmetrical auxin transport. PIN4 in the quiescent center mediates auxin efflux towards the columella cells [32]. The auxin sink in the root cap contributes to the asymmetric auxin distribution in reoriented roots [33]. Polar localization of PIN2 in the plasma membrane of the lateral root cap and epidermal cells directs the auxin flow toward the cortical and epidermal cells [34,35]. Any auxin asymmetry leads to differential growth in the elongation zone [36].

In hypocotyls, sedimenting amyloplasts are found in the endodermis [37,38], which acts as gravisensing layer in dicots and is essential for the graviresponse [39,40,41]. PIN3 is expressed in the endodermis [28] and localizes laterally after two minutes of gravistimulation [42]. PIN3 localization is auxin dependent [43] and a mutation in the *PIN3* gene affects differential growth [42]. Hence, *PIN3* is an important factor for shoot graviresponse [43]. The *PIN3* homologs *PIN4* and *PIN7* are not expressed in the shoot endodermis and their mutants do not affect the shoot graviresponse [43]. Along with PIN proteins, Altered Response to Gravity (ARG1) is required for the localization of PIN3 during the early stages of hypocotyl graviresponse [44]. A mutation in *ARG1* affects early gravisignaling in roots and hypocotyls [45] but can be corrected by the expression of ARG1 in the endodermis of hypocotyls and root caps, which implies that it functions in statocytes [46]. PIN1 re-localization alters the lateral redistribution of auxin and affects the phototropic [47], and gravitropic response [48] and *PIN1* mutants show reduced polar auxin transport (PAT) in *Arabidopsis thaliana* inflorescence axes [49]. Another Auxin Response Element promotes *IAA5*, a member of Aux/IAA gene family, which has been identified as one of the early and primary response genes [50,51] that are governed by auxin [52]. 

Based on these auxin networks, we chose to compare the transcription levels of *PIN1*, *PIN3*, *IAA5* and *ARG* in vertically and horizontally reoriented hypocotyls of *Brassica rapa*. Since Ubiquitin (UBQ) is highly expressed [53] and used as a reference gene [54,55,56], its transcription level was used as a reference to assess changes in the transcription of auxin responsive genes. 

Despite many studies on PAT in hypocotyls and on effects of auxin on gene expression [57,58] during graviresponse [59], spatio-temporal differences in gene regulation during hypocotyl curvature have not been studied. Typically, total RNA is extracted by pooling the samples from the same region from several seedlings, which prevents the assessment of gene transcription at precise locations in the tissue/region under consideration. High spatial resolution is desirable as size, age and development is likely to affect transcription and expression profiles [60]. In addition, auxin-induced transcription is likely to follow a temporal profile that cannot be assessed by destructive sampling methods. We used Solid Phase Gene Extraction (SPGE) [59,61], which relies on mRNA binding to functionalized probes and allows for the repeated sampling of tissue. The temporal and spatial changes of transcription profiles during the initial hypocotyl curvature indicate an auxin-based regulation of positive and negative hypocotyl curvature.

## 2. Results

### 2.1. Time Course of Curvature

As reported earlier [1], dicotyledonous plants such as Brassica, Radish and Flax show a bi-phasic and combined graviresponse of the root-shoot axis. The time course of curvature involves a transient curvature of the root tip. Upon straightening of the tip, the hypocotyl base begins to respond as positively gravitropic. The graviresponse is completed after about 20 h. The sampling intervals were chosen to coincide with the onset of hypocotyl curvature (four hours after reorientation), the midpoint of curvature (45-degree curvature of the hypocotyl; 11 h) and vertically oriented seedlings (48 h after the initial reorientation).

### 2.2. Nucleic Acid Extraction, Yield, and Reference

RNA extraction of tissue was performed along the upper and lower flank of the basal hypocotyl from tissue extraction or by Solid Phase Gene Extraction (Figure 1).

The average cDNA yield was based on the amount of cDNA after reverse transcription. The tissue extraction (258 ± 16 µg) was significantly higher (*p* < 10^−10^) than the yield from SPGE probes (0.654 ± 0.024 µg). Nonetheless, reliable qPCR experiments of SPGE samples were possible. Although UBQ1 transcription was not steady (Figure 2), the changes were small in comparison to auxin-sensitive genes and UBQ1 transcription between the upper and lower flank remained similar (*p* = 0.75) throughout the experimental period (Figure 3); therefore, UBQ1 is an acceptable reference gene.

### 2.3. Transcription in the Hypocotyl Base Tissue

The transcription of auxin related genes PIN1, PIN3, IAA5 and ARG1 were similar on the left and right flanks of vertically oriented hypocotyls (Figure 4) but changed after horizontal reorientation. PIN3, IAA5 and ARG1 were downregulated along the top flank but upregulated in the bottom tissue. PIN1 transcription did not change during the first four hours but after 11 h of reorientation increased in the lower flank and decreased along the upper flank. At that time, all genes showed maximal changes in transcription activities (Figure 5).

The transcription analysis by SPGE showed similar profiles as the tissue data. The downregulation of PIN1, PIN3, IAA5 and ARG1 along the top coincided with upregulation along the lower flank (Figure 6) four and 11 h after reorientation. After 48 h, when the hypocotyl approaches vertical position, this trend of transcription expression was reversed, i.e., transcription was upregulated in the upper flank and downregulated in the lower flanks.

SPGE sampling showed the largest difference in transcription at the point of the maximum curvature (probe #2 and #5, Figure 1) compared to the adjacent positions after four and 11 h. The ratio of the fold change over time of all genes at position five vs. two varied by gene and showed the smallest signal after 48 h (Figure 6). Considering the transcription for probe two and five (the points of maximal curvature on the convex and concave flank, respectively, for different time points showed that after four hours, the ratio for PIN1 and ARG1 was similar. IAA5 transcription was twice as high at the bottom (#5) than top (#2) position. At 11 h, PIN1, PIN3, and ARG1 showed similar values for the convex and concave sites. The polarity of IAA5 was less, but still reached about five times the value of the top flank. After 48 h, the values were reversed and the polarity for PIN1, PIN3, and ARG1 between the top and bottom side was about six to eight-fold; the polarity for IAA5 was weaker. These data show that after reversal of the curvature the transcription of the examined genes reached a comparable but reversed polarity as after 11 h.

### 2.4. Tissue and SPGE Extraction

The average of the transcription values of all three SPGE probes on either flank was compared with corresponding tissue data of the lower and upper flank. The SPGE values for all examined genes were similar or slightly higher than the tissue values. The overall profile for all tested genes was comparable for both methods, i.e., upregulation along the bottom flank and downregulation in the upper side until 11 h after reorientation (Figure 7). Consistently, this trend was reversed after 48 h. The transcription of all genes showed the greatest variability 11 h after reorientation, consistent with the onset of curvature reversal from positive to negative.

### 2.5. Comparison of the Hypocotyl and Root Tip

As illustrated [1], the hypocotyl responds to gravity in sequentially positive and then negative curvature. In contrast, the root tip always exhibits positive gravicurvature. Therefore, a comparison of both tissues shows the extent and magnitude of synchronous changes in the transcription of relevant genes.

ARG1 showed the highest transcription value compared to the other examined genes (Figure 8). The transcription of all genes increased strongly at four hours followed by a gradual increase at 11 h after reorientation. SPGE showed upregulation for both PIN1 and ARG1 at all times after reorientation. PIN3 transcription remained constant until four hours after reorientation, followed by continued but reduced upregulation. Although IAA5 increased in the tissue (red line) this gene remained unchanged for SPGE data. Interestingly, the magnitude of upregulation for all genes at 11 h after SPGE extraction was in the range of the earlier tissue data points, except for ARG1. Therefore, a drastic difference in values was observed. Similar to the root tip, we considered the average of the hypocotyl tissue (i.e., the average of top and bottom flanks) and found that the transcription levels of all genes were similar (*p* = 0.25) for the hypocotyl and root tip regardless of orientation or time after reorientation.

## 3. Discussion

### 3.1. UBQ1 as Reference

We verified the usage of UBQ1 as a reference, as UBQ1 expression in both sides of the vertical and top and bottom flanks of the curving hypocotyls did not change significantly (Figure 2). The observed, minor changes in UBQ1 may not be related to curvature but the effect of development and/or differentiation. Hence, UBQ1 was identified as a reference gene [62] and it was equally expressed in the upper and lower flanks of the gravicurving roots of the Brassica rapa seedlings [59].

### 3.2. Transcription of Auxin Related Genes

The transcription of PIN1, PIN3, IAA5 and ARG1 varied during the graviresponse of *B. rapa* and is likely mediated by auxin as auxin-induced elongation along the convex side of hypocotyls coincides with the upregulation of these genes, while the concave side shows the reduction in transcription. The connection of these genes with auxin is well established [29,63,64], and at least for PIN1 and PIN3, the transcription pattern over time was identical (Figure 5). The initiation and termination of bending [65] is in line with our observation that the hypocotyl cortex that was probed by SPGE and shows the PIN1 and PIN3 transcription pattern that initially activates the auxin flow to the upper flank, but eventually reverses its lateral polarity to account for the upward orientation of the hypocotyl [1]. The differential elongation [4] corresponds with the observed transcription changes and indicates the dependency of basal hypocotyl curvature on auxin mediated processes. The consistent pattern of initial upregulation along the bottom flank and eventual reversal contradicts the notion that hypocotyls curve negatively gravitropic but are in line with earlier observations that young basal hypocotyls of *B. rapa*, Linum usitatissimum and other dicots behave similarly to roots and bend positively gravitropic during early growth periods. This behavior reverses as the hypocotyl matures and the orientation of the primary root has been established [1].

### 3.3. Spatio-Temporal Changes during Hypocotyl Curvature

The location of tissue and progression of curvature affect gene expression, repressors, and other auxin signaling factors [66]. The varying transcription of genes in *B. rapa* roots [53], indicates transcription differences along the curving hypocotyl; their variance over time should coincide with temporal and spatial changes of curvature and related gene expression. All examined genes exhibited changes in their transcription in the upper and lower flank (Figure 3 and Figure 5, Figure 6 and Figure 7) of gravicurving hypocotyl over time. The initial upregulation in the lower flank and downregulation along the upper flank is in line with the observed downward curvature in young hypocotyls [1]. The increased transcription activity along the lower flank is indicative of auxin-induced growth inhibition.

As the downregulation of all genes on the upper flank at the beginning of hypocotyl curvature (four hours) is greater than the upregulation on the bottom side, the differential ratio (up/down, Figure 5 and Figure 6) is high and suggests auxin flow to the lower flank. At the midpoint of hypocotyl curvature at 11 h [1], the auxin asymmetry reaches its maximum and is likely associated with the tipping point [3] that leads to the subsequent reversal of curvature at about 35 h for PIN1 and IAA5, and 30 h for PIN3 and ARG1. This sequence coincides with the reversal of hypocotyl downward curvature. The transition in gene transcription implies a shift in either lateral auxin transport or auxin sensitivity, possibly mediated by reactive oxygen species or reductive factors [67]. The significance of this observation is the ensuing change of curvature from positive to negative gravitropic and that it reaches the typical negative gravitropic curvature that is characteristic of mature hypocotyls.

Although the transcription profile 48 h after reorientation was not equal on both flanks, the data suggest that the hypocotyl curvature for the different curvatures rely on the same genes but does not address what causes the reversal of auxin sensitivity over time. However, the effect of auxin levels and ensuing gene transcription will also be affected by the cessation of the ability to elongate; therefore, the maturing tissue no longer responds to changes in auxin-mediated development and the reversal in hypocotyl curvature, i.e., negative gravitropism, depends on tissue that is still expanding.

The high-resolution SPGE analysis revealed changes in the transcription profile within the curving tissue (Figure 6). The greatest polarity was associated with the central probes (#2 and #5), therefore SPGE profiling is sensitive to spatial differentiation and expansion. The differences between the central and distal probes illustrate both auxin effect and the spatial dependency of gene activation. Future work will investigate if vertically oriented hypocotyls show maturation-dependent changes in gene transcription. The combination of sensitivity, age, and stimulation (auxin) are likely to control (differential) elongation 

Despite the overall similarity in the assessment of transcription activities, there were differences between the individual genes and sampling positions. Notably PIN1 and PIN3 showed its lowest transcription after four hours at the apical-most upper position and increased without reaching a minimum at 11 h. The same trend prevailed for IAA5 and ARG1 (Figure 6). Possibly, the apical region is less sensitive for curvature, and thus, inhibition. Another consideration relates to the expression of PIN3 and ARG1 in the endodermis [46] and their involvement in shoot gravitropism [43] via polar auxin transport. Therefore, the signal may not have been maximal as SPGE sampling clearly involved cortical tissue. Nonetheless, the polarity of all examined genes was higher for SPGE than for tissue analyses (Figure 5 vs. Figure 6), supporting SPGE-based sampling as a valuable tool for the study of physiological processes.

IAA5 is considered a gravitropic response indicator as its distribution reflects gravity-induced auxin asymmetry [68]. IAA5 had the highest transcription levels (lowest Cq values, Figure 4). However, its lateral polarity was less than that of the other examined genes. 

### 3.4. Comparison of Root Tip and Hypocotyl

The root tips of horizontally oriented seedlings reach a vertical position after four hours of reorientation [1]. Therefore, lateral auxin gradients are likely to not exist at longer times. The comparison between root tip tissue and SPGE sampling shows that both sampling methods detect a similar change in transcription, but tissue samples show larger signals during curvature. Since no concave or convex tissues were sampled, the signal likely reflects a general increase in transcription. SPGE shows a similar increase, but after a significant delay. Both sampling profiles showed a comparable order of activation. The notable exception was IAA5, which was substantially higher in the tissue and remained unchanged in SPGE samples. This discrepancy may stem from a higher mRNA quantity in tissue that was not represented in SPGE extracts such as the developing stele or epidermis [69]. The rapid increase in all transcripts in tissue samples occurred over a longer time in SPGE samples. This increase is not related to curvature but likely illustrates the increase in transcription as the root ages or sensitivities change during vertical orientation. Changes in sensitivity with age have been documented previously [1,70,71] and indicate that despite the lack of curvature-specific differences, root tips are a suitable reference for hypocotyl-related changes in gene transcription. 

This notion likely extends to other species that show a biphasic response in hypocotyl curvature. Equally important is the confirmation that the positive and negative hypocotyl curvature depends on similar hormonal signaling mechanisms as root curvature.

### 3.5. Comparison of RNA Extraction Techniques

Although the yield of tissue extraction is about 400 times the yield of SPGE extraction (see Section 2.1), SPGE data show greater polarity than tissue extraction (Figure 5, Figure 6 and Figure 7). The large tissue quantity not only diminishes the differential signal but is problematic as large quantities make differential sampling unreliable. The much higher spatial resolution of SPGE avoids such complications, is accurate, and can readily and repeatedly determine RNA profiles without purification [61]. Overall, the data show that SPGE allows for mRNA profiling at high spatial resolution that is useful for the determination of transcriptional responses to physiological stimuli in plants and other biological systems.

## 4. Materials and Methods

### 4.1. Plant Material

Seeds of Brassica rapa vs. rapa were germinated at 25 °C in darkness between germination paper that was positioned on top of five mm thick floral foam to avoid thigmotropic stimulation of roots upon emergence. Seedlings with an average root length of 10 mm were used for experimentation 42 h after imbibition. All experiments were performed in moist petri dishes.

### 4.2. Gravistimulation and Tissue Sampling

Vertically grown and positioned seedlings were reoriented such that the entire seedling was positioned horizontally. Before reorientation and 4 and 11 h after reorientation, the hypocotyl was dissected into two parts (left/right for vertical controls and upper/lower for horizontally oriented seedlings). Samples obtained after 48 h did not include tissue samples as the tissue size after 48 h of additional growth exceeded the extraction capacity. The apical two mm of the root tip were removed using a scalpel and used as additional reference tissue. All tissue samples were immediately submersed in liquid nitrogen.

### 4.3. RNA Extraction

Total RNA extraction of tissue (hypocotyl and 2 mm long root tip) was performed using the Sigma Spectrum™ Plant Total RNA Kit. For each time point three biological replicates with material from 20 seedlings each were analyzed. The samples were homogenized in a microfuge tube for one min in lysis solution followed by vortexing, then heated (50 °C for 5 min in a water bath) and centrifuged (16,000× *g*, 4 min). The supernatant was transferred into a microfuge tube containing a filter column and centrifuged (16,000× *g*, 90 s). The filtrate was transferred to a binding column and centrifuged at 16,000× *g* for 90 s. The flow through, as well as the solution from three successive wash steps (500 µL wash buffer each), were discarded. The filter column was centrifuged to remove any remaining solution before the elution buffer was added for 1 min, followed by centrifugation (16,000× *g* for 90 s). The eluant contained total RNA and was immediately put on ice. The RNA was quantified using a DeNovix DS-11 spectrophotometer. The total RNA was diluted with deionized DEPC water to a concentration of 100 ng/µL.

### 4.4. SPGE Probe Preparation mRNA Extraction

Solid Phase Gene Extraction used #6 acupuncture needles (SEIRIN, Kyoto, Japan). Batches of 40 needles were sonicated sequentially in hexane, acetone, and ethanol for 15 min each, followed by air drying at 80 °C for 15 min. The needles were functionalized by incubation in a mixture of 0.25 mL (3-glycodoxypropyl) trimethoxy silane (3GTMO, SPI-Chem, West Chester, PA, USA) in 0.75 mL xylene and 10 µL diisopropylethylamine in a sealed amber glass vial at 80 °C overnight. The needles were cooled, washed thrice with ethyl acetate, air dried for 10 min and incubated in 1 µM NH2-oligo (dT)15 nucleotides in 0.1 M KOH at 37 °C for 6 h (four needles per 20 µL solution in closed PCR tubes). The functionalized needles were rinsed thrice with deionized water (50 °C) for 5 min each, followed by drying at room temperature. Functionalized needles were inserted for one minute into the hypocotyl and the root tip (Figure 1) four, 11 and 48 h after reorientation. The probes were manually inserted in about 10 s intervals about one mm deep into the cortical tissue in the sequence shown in Figure 1. Hypocotyls were discarded if the probes were placed inaccurately. After 11 h, the curvature reached about 45 degrees, which resulted in stacked arrangement of the loop ends of the probes. Root tips were sampled by inserting the needle into the approximate area of the quiescent center. This approach was chosen as this tissue is not differentiated. For each time point, needles from three different seedlings were combined according to their position and considered one biological sample. Three biological samples (i.e., nine seedlings) were measured.

### 4.5. Reverse Transcription

Two µL of oligo-dTs and RNA (10 µL of total RNA or three SPGE probes) were added to a PCR tube. The tubes were heated for five min in a Thermocycler (Hybaid, PCR Express) to 80 °C. A master-mix was prepared that contained 4.2 µL deionized water for total RNA or 16.2 µL for SPGE samples. All samples contained 2 µL 10 X buffer, 0.8 µL 25 mM dNTP and 1 µL reverse transcriptase (final concentrations: 1 mM dNTP, 1X reverse transcription buffer, and 50 U Multiscribe^TM^ reverse transcriptase (Applied Biosystem high-capacity cDNA kit). This master-mix was added to RNA/oligo dT mix (eight µL for total RNA and 20 µL for SPGE) and incubated at 25 °C for 10 min, then at 37 °C for 2 h, and deactivated at 85 °C for 5 min. cDNA was quantified using a DeNovix DS–11 spectrophotometer.

### 4.6. Transcript Quantification Using qPCR

cDNA was diluted with diethyl pyrocarbonate (DEPC) treated water to a concentration of 250 ng/µL and stored at −20 °C. The transcripts of UBQ1, PIN1, PIN3, IAA5, and ARG1 were measured using qPCR with primers and their efficiencies listed in Table 1. Each sample consisted of three biological and three technical replicates each. Each 10 µL reaction contained 5 µL 2X SYBR green PCR mastermix, 1 µL primer mix (0.5 µL forward +0.5 µL reverse primer, 250 nM final concentration) and 4 µL of 250 ng/µL cDNA. qPCR was started at 95 °C and then cycled 40 times between 60 °C (60 s) and 95 °C (15 s) in a StepOne^TM^ qPCR system (Thermo Fisher Scientific, Waltham, MA 02454, USA). All qPCR products were subjected to melt analysis.

### 4.7. Analyses

The fold-changes in transcripts for different times of reorientation were referenced to UBQ using the ΔΔ Cq method [72]. Log-transformed data avoid skewness of small values and results in symmetrical representation of up- and downregulation of transcription [73]. The ratios of Cq values of UBQ1 obtained from bottom vs. top flank or for SPGE needle 1, 2, 3 (=top or left) and needle 4, 5, 6 (=bottom or right) were calculated for each condition and used as reference. All Cq values, ratios, or log (2) data transformation was performed in Microsoft Excel and statistically analyzed using the ANOVA function (Excel 2019).

## 5. Conclusions

Our results demonstrate that the basal hypocotyl curvature depends on auxin mediated process and involves the same genes for downward or upward curvature that were found to be involved in root curvature. The expression trend for auxin related genes in positively gravitropic hypocotyl curvature reverses when the hypocotyls begin to develop negative gravitropism. High resolution sampling revealed spatial variation in the transcription of auxin related genes throughout the curvature with the greatest polarity present at the most concave and convex position in the curvature.

## Figures and Tables

**Figure 1 plants-11-01191-f001:**
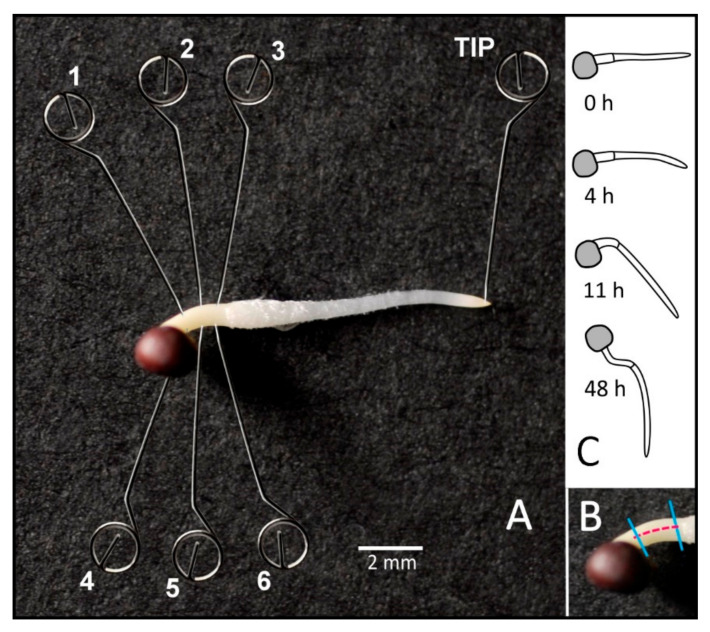
(**A**) Extraction of mRNA using Solid Phase Gene Extraction: Oligo-dT15 functionalized stainless-steel needles were inserted into the upper and lower side of horizontally placed hypocotyls and the root tip of Brassica rapa seedlings. The hybridization between the dT and poly A tail of the mRNA was accomplished in 60 s. (**B**) Drawings that represent hypocotyl curvature during the experiments. (**C**) Sampling of the upper and lower flank of hypocotyl tissue.

**Figure 2 plants-11-01191-f002:**
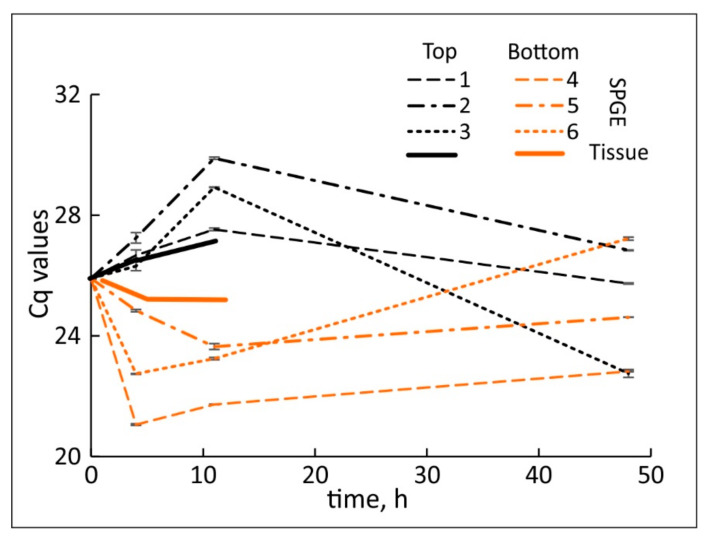
Absolute Cq values for UBQ1 obtained by tissue extraction (solid lines) and SPGE (patterned lines). The top flank showed reduced (higher Cq values) transcription compared to the bottom flank. The numbers 1–6 correspond to locations in Figure 1. Mean ± SE, *n* = 9.

**Figure 3 plants-11-01191-f003:**
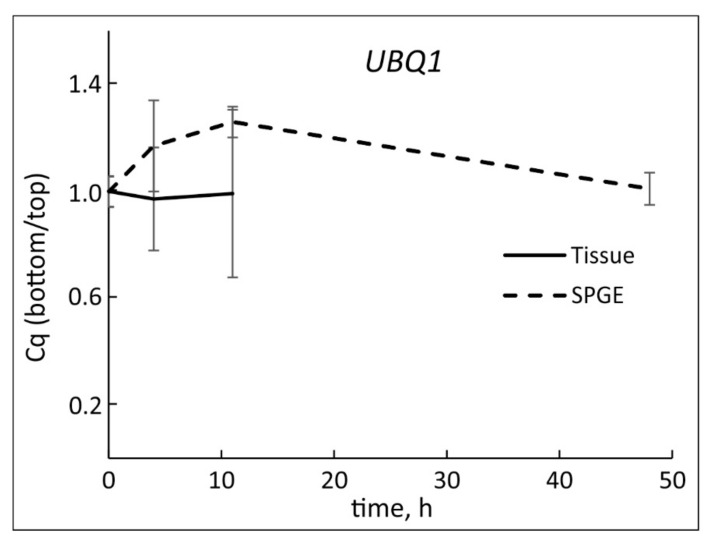
The ratio of Cq values of UBQ1 in bottom vs. top flank obtained from hypocotyl tissue (solid line) and SPGE (dashed line) after horizontal reorientation of *B. rapa* seedlings. Time = 0 refers to vertical position. Transcription of UBQ1 did not change significantly after reorientation (*p* = 0.75); mean ± SE, *n* = 9.

**Figure 4 plants-11-01191-f004:**
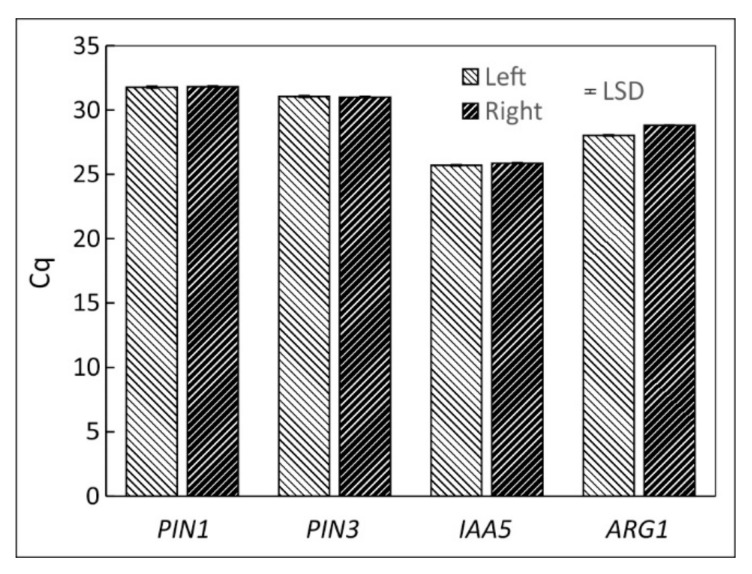
Absolute Cq values of the investigated genes PIN1, PIN3, IAA5 and ARG1 obtained after extracting tissue of the left and right flank of a vertically grown *B. rapa* seedlings. Mean ± SE, *n* = 9.

**Figure 5 plants-11-01191-f005:**
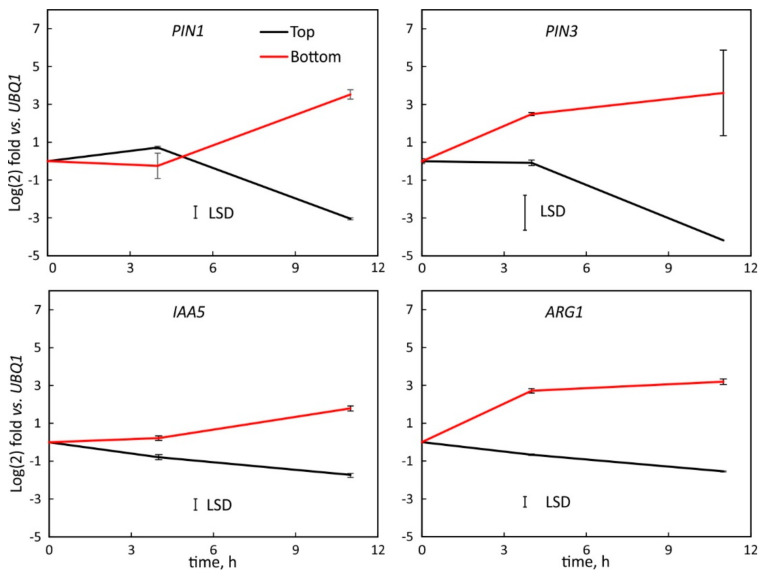
Log (2) fold change in transcription level of PIN1, PIN3, IAA5 and ARG1 in tissue samples of the upper (black) and lower (red) flank of hypocotyls of horizontally reoriented Brassica rapa seedlings four and 11 h after reorientation relative to the transcription of UBQ1. The transcription was upregulated in the bottom flank and downregulated in upper flank. The least significant difference (LSD) is based on the error of all measurements per panel. Mean ± SE, *n* = 9.

**Figure 6 plants-11-01191-f006:**
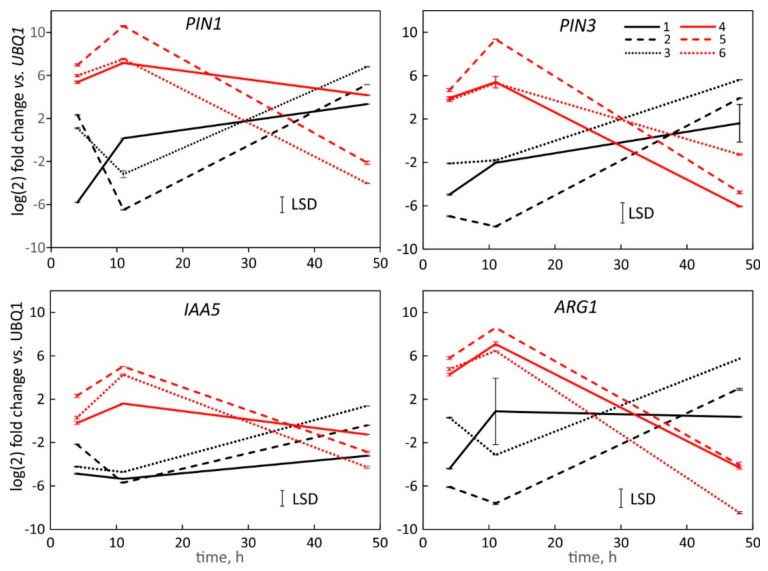
Log (2) fold change in transcription level of PIN1, PIN3, IAA5 and ARG1 after Solid Phase Gene Extraction (SPGE) of curving Brassica rapa hypocotyls relative to UBQ1 values. Probe 1 (apical end of curvature zone), 2 (center), and 3 (basal end, at root/shoot junction) sampled the upper flank (black lines) and probes 4, 5 and 6 sampled the corresponding positions along the bottom flank (red lines), see Figure 1. The transcription profiles are shown for 4, 11 and 48 h after horizontal reorientation. Positive and negative values indicate up- and downregulation, respectively. The least significant difference (LSD) is based on the error of all measurements per panel. Mean ± SE, *n* = 9.

**Figure 7 plants-11-01191-f007:**
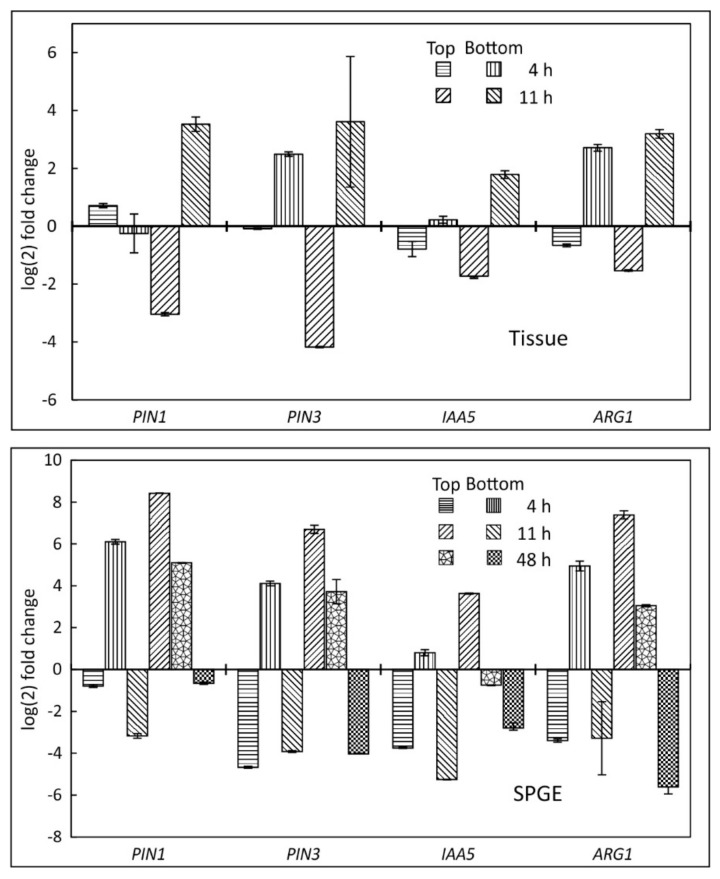
Comparison of transcription profiles of PIN1, PIN3, IAA5 and ARG1 in *B. rapa* seedlings in tissue after four and 11 h of reorientation (upper panel) and SPGE extraction (lower panel) after four, 11 and 48 h reorientation. The data for SPGE top and bottom refers to the average values of the combined probes 1, 2, 3 and 4, 5, 6, respectively. All values are shown as fold change based on UBQ1. Mean ± SE, *n* = 9.

**Figure 8 plants-11-01191-f008:**
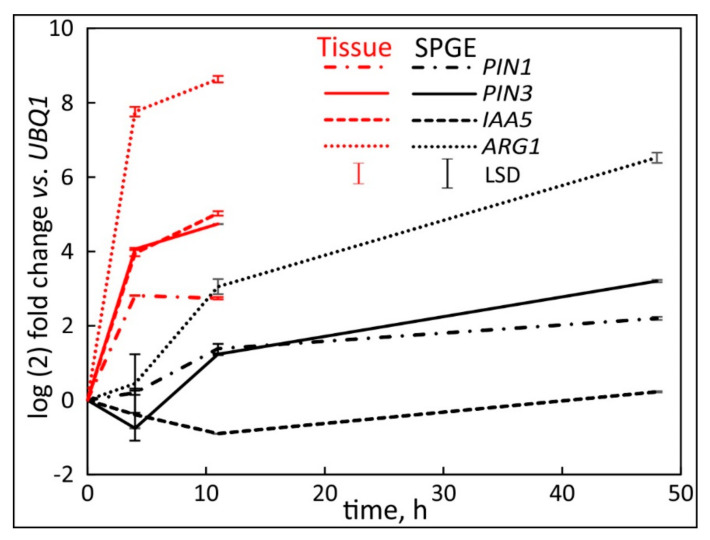
The transcription profile of PIN1, PIN3, IAA5 and ARG1 in the root tip after horizontal reorientation of *B. rapa* seedlings. Data were tissue (red) or SPGE (black) sampling of the 2 mm tip region. The transcripts were referenced to UBQ1. The least significant difference (LSD) was calculated separately for tissue and SPGE samples. Mean ± SE, *n* = 3.

**Table 1 plants-11-01191-t001:** Primer information for UBQ1, PIN1, PIN3, IAA5 and ARG1. The amplicon size includes the primer pairs.

Gene (Accession #)	Sequence (5′–3′)	Tm (°C)	Amplicon (bp)	Efficiency (%) *
*UBQ1* (Z24738.1)	F: GGAGAGCAGTGACACCATCGA	58	120	103.1
	R: GCCAAGGTACGACCATCTTCA			
*PIN1* (AJ132363.1)	F: ATCTTCACACCGACGGTAAGTC	54	155	111.5
	R: GTCAGATCTTCCACCCTGTTCA			
*PIN3* (AJ249298.1)	F: TCTTAACGTTTCCGATGGAGCC	58	167	103.2
	R: CTCTTCCAGCGAAACTAAACCG			
*IAA5* (NM101427.4)	F: GCATGGATGGAGCTCCTTATCT	58	171	96.5
	R: GCATCCAATCTCCATCTTTGTC			
*ARG1* (AT1G68370.1)	F: TTTGGCGCGGTTTCAAGAAG	54	120	95.3
	R: CCGCTTGGTGTCTTCACAAC			

* Based on the slope of serially diluted (1:10, 1:100, 1:1000 and 1:10,000) measurements with three technical replicates each.

## Data Availability

Data are available upon request.

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
