# Peer review of "Transcription Profile of Auxin Related Genes during Positively Gravitropic Hypocotyl Curvature of *Brassica rapa"

_plants, 2022, doi:10.3390/plants11091191_

Round 1
Reviewer 1 Report
MS titled "Transcription Profile of Auxin Related Genes during positively
gravitropic hypocotyl curvature of Brassica Rapa" is actually methodological article comparing two approuches in transcript level analysis in Chinese cabbage seedlings upon curvature. I think it should be pointed in title. Ms is confusing and need to be clarify. Fig 1 needs to be positioned first, not last. I suggest to add seedlings and sampling photo of "clasical method" on Fig 1. The same results are repeted on different figures, so I suggest to avoid that. MS is acctually more appropriate for Short communication, rtaher than full article. Some figures such as Fig 2, 3, 4 may be replaced in suppl materials.
My oppinion is that MS is not appropriate for publication and need to be revised before consideration for publication in Plants.

Author Response
Reviewer 1:
States that the article compares two approaches of transcript level analysis during curvature and wants that to be included in the title.
The current title (Transcription profile of auxin related genes during positively
gravitropic hypocotyl curvature of Brassica Rapa) emphasizes the biological mechanism of the initially positive gravitropic curvature rather than a methodological approach. While the SPGE-based assessment is essential for the spatial analysis, it is not the main topic of the paper. To not complicate the title further we (and apparently the other reviewers as well) would rather keep the current title.
Points out that Figure 1 is not positioned at the beginning although it is referenced first. This arrangement stems from the journal’s stipulation to place material and methods AFTER the results. To correct the sequence, we have added an introductory sentence to the section 2.1 and placed Figure 1 in this section. The added text reads: RNA extraction of tissue was performed along the upper and lower flank of the basal hypocotyl from tissue extraction or by Solid Phase Gene Extraction (Figure 1).
Wants a sampling photo of the classical method. We changed Figure 1 to include two additional items, a sketch that shows tissue sampling (Fig. 1B) and drawings for the time course of the seedling curvature (Fig. 1C).
Complains that the same results are repeated several times. We have carefully chosen the current type of illustration to emphasize respective points. The representation of raw Cq values is different from data calibrated against UBQ1 transcription.
Suggests treating Figures 2, 3, 4 as supplementary material. We feel that these figures are essential for the understanding of the work and the nuanced illustration of results and decided to keep these figures.
Reviewer 2 Report
The manuscript by Chitra Ajala and Karl Hasenstein (PLANTS- 1672327: "Transcription Profile of Auxi-Related Genes During Positively Gravitropic Hypocotyl Curvature of Brassica rapa") reports on the results of targeted experiments aimed to investigate the levels of transcripts derived from auxin-related genes (including PIN1, PIN3, IAA5 and ARG1) in upper and lower flanks of gravistimulated young Brassica rapa seedlings using a combination of tissue-level RNA extraction and Solid Phase Gene Extraction (a novel method that relies on mRNA binding to functionalized probes and allows repeated sampling of very well defined regions of a plant organ at multiple phases of a response), both coupled with qPCR analysis. The authors investigate the hypocotyl gravitropic response of young Brassica rapa seedlings, which they previously showed to initiate a first phase of positive (downward) bending, subsequently followed by a normally negative gravitropic response. They show that both methods lead to similar results, showing increased expression of these auxin-related genes on the bottom flank at the basal side of the hypocotyls, and decreased expression on the top flank, during the initial phase of downward curvature. They also show this initial phase is followed by opposite trends during the second phase involving negative gravitropism. The SPGE method provided sufficient spatiotemporal resolution to demonstrate the central region of the probed area, which was chosen because it displays the largest contribution to gravicurvature, also showed the largest difference in gene expression between the two flanks. Furthermore, the largest differential expression between lower and upper flanks in that region coincided with the 4h and 11h time-periods. The authors conclude that the transcriptional response of these auxin-related genes at this early phase of gravitropism is compatible with the opposite curvatures displayed by the hypocotyls at the corresponding phases of the response, from positive to negative (20h and more).
The existence of this surprising phase of positive gravitropism preceding the normally negatively gravitropic phase at the base of the hypocotyls of most dicotyledon seedlings was only recently discovered by this same group. This report provides an interesting expansion on these observations, demonstrating differential expression of auxin-related genes upon gravitropism during these phases of the response. While a global transcriptome analysis using similar plant materials would have been entirely feasible and would have provided much more information on the molecular mechanisms that accompany the phases of positive hypocotyl gravitropism in young Brassica seedlings followed by curvature reversal, the present report is still of interest as it characterizes the transcriptional response of several auxin-related genes at different phases and locales along the base of gravistimulated hypocotyls early in response to gravistimulation. As such, I believe it is worthwhile information. This being said, I have several comments and suggestions the authors may want to consider in order to potentially improve the quality of their manuscript.
1) The Materials and Methods are rather cryptic with regard to the SPGE extraction method. Information that would be useful here includes the depth of penetration of the probes within the responding tissues (does the probe penetrate at depths beyond the epidermis, into the cortex and/or endodermis [particularly relevant for ARG1 and PIN3]? How many tissue layers are crossed with the probe? Are we looking at joint expression within all penetrated cells (going inward), or is the epidermis mostly represented (for instance if mRNA binding saturation occurs as the probe progresses through the epidermis)?)
2) Also, section 2.1 describes the time course of hypocotyl curvature response to gravistimulation and precisely defines the curvature displayed by hypocotyls at the harvest times chosen in this study. However, the authors do not seem to include the second phase of negative gravicurvature that follows the initial downward response. The last time point corresponds to when the hypocotyl is now back to the vertical. Yet, the abstract suggests the up-down expression gradient reversal observed in their work corresponds to the reversal of curvature response from negative to positive gravitropism. Paragraph 2.1 should more clearly define the relationship between time points and hypocotyl curvatures in this work. Adding panels to Figure 1 illustrating the various phases of the seedling response to gravistimulation at the sampling time points would actually be very helpful.
3) Still related to point 2 above, is there any information on the extent (and sign) of curvature development within the seedlings subjected to this analysis? How much variability exists between seedlings, and were the probed sites chosen based on average information on spatial distribution of curvature within the population of seedlings, or was it tailored to individual seedlings based on their curvature responses?
4) Were the probes inserted sequentially in the various chosen sites (#s 1 through 6)? If they were, how much time elapsed between successive probings? Considering the likely mechanical/wounding impact tissue puncturing may have on the physiological status of plant organs and their transcriptome profiles, it seems important to provide additional information that would allow any lab to recapitulate the same experimental set up to verify the data, interpret them and possibly extrapolate to additional, more complete inquiries of this process.
5). Some imprecisions can be found throughout this manuscript. For instance, the authors seem to use "cortical" to more generally define peripheral tissues, rather than using the more common definition attributing this name to some of the cell layers located between the epidermis and the pericycle. This becomes important when describing the pattern of PIN2 expression in the root tip, for instance (see the Introduction, page 2, lines 49, 58). I suggest using the words "cortex" or "cortical" only within the context of the tissue type.
6) In the introduction, the authors suggest PIN4 is specifically expressed in the quiescent central where it serves to transport auxin from the provasculature into the columella region of the cap (page 2, line 69). In fact, PIN4 expression encompasses several additional cell types in the root tip, above the quiescent center, where they target the transport of auxin toward the quiescent center and the columella region of the cap.
7). As already discussed in point 1 above, the data presented in this manuscript are rather simple (expression levels defined by qPCR (relative to a UBQ control) of 4 genes in different regions of the hypocotyl base upon gravistimulation). Yet, the authors' presentation of the data seems unnecessarily complicated. Examples of concepts I had difficulty understanding follow:
- Why are the data presented in the form of logarithms of different bases (ln in Figure 7 vs base-2 log for all other figures)? Shouldn't the same transformation be used everywhere?
- I simply do not understand the second half of the last paragraph of section 2.3 (page 6, lines 169-176). For instance, the two sentences in lines 169-173 seem wrong based on a simple analysis of Figure 6. Am I missing something here?
- The legend to Figure 7, lower graph, should specify that these data correspond to average values for the three probes on each hypocotyl flank.
8) Symbols indicating statistical significance should be added to each graph. This would help interpretation of the results.
9) In Materials and Methods, which cv./variety of Brassica rapa was used in these experiments?
9) The discussion should probably also include a paragraph discussing the fact that differential expression of auxin-response genes at the early phases of downward hypocotyl curvature followed by upward reversal, is compatible with changes discussed in the root tip (auxin-responding AUX/IAA genes being upregulated on the concave side during graviresponse), not with those expected in older hypocotyls exposed to gravistimulation (where auxin promotes elongation on the convex side, at least for the most apical regions of the hypocotyls).
Author Response
Reviewer 2:
Complained that Materials and Methods are rather cryptic with regard to the SPGE extraction method. We provide the insertion details and point out that the cortical tissue of the hypocotyls was sampled (new L379 ff). During the penetration process the mRNA binding and extraction are minimal. The actual hybridization occurs during the static 1-minute extraction phase. Since these technical aspects were reported earlier (BioTechniques 2008, 45, 172-178) we did not repeat these details.
Requested defining the relationship between time points and hypocotyl curvatures. We added additional details to the material and method section and expanded Figure 1 to illustrate sampling, stages of curvature, and tissue sampling.
Asked about the extent of curvature development within the seedlings subjected to this analysis. We added the serial drawings depicting various stages of curvature as part of Figure 1. Of course, there was (biological) variability, but we selected hypocotyls that were close to the previously described phenotype (Plant Science 2019, 214-223).
Inquired about the sampling process such as whether the probes were inserted sequentially. That information is detailed in L381. The insertion reads: “The probes were manually inserted in about 10 second intervals about one mm deep into the cortical tissue in the sequence shown in Figure 1. Hypocotyls were discarded if the probes were placed inaccurately. After 11 hours, the curvature reached about 45 degrees, which resulted in stacked arrangement of the loop ends of the probes. Root tips were sampled by inserting the needle into the approximate area of the quiescent center. This approach was chosen because this tissue is not differentiated.” This information should be sufficient for anyone to replicate the experiments.
Pointed out some inconsistencies regarding the term cortex. We are referring to cortex in the introduction as general flow pattern, not as specific for PIN expression
Asked to correct the PIN4 expression pattern. While the concern is appreciated, we do not state the claimed fact anywhere. Therefore, a correction does not apply.
Wondered about the use of ln in figure 7 vs. log(2) elsewhere. The ln formulation was incorrect and unintended (a typo) and has been corrected.
Indicated that the text in L 169-176 was difficult to understand. We have corrected the text and it now reads: SPGE sampling showed the largest difference in transcription at the point of the maxi-mum curvature (probe #2 and #5, Fig. 1) compared to the adjacent positions after four and 11 h. The ratio of the fold change over time of all genes at position 5 vs 2 varied by gene and showed the smallest signal after 48 h (Fig. 6). Considering the transcription for probe 2 and 5 (the points of maximal curvature on the convex and concave flank, respectively, for different time points showed that after 4 h, the ratio for PIN1 and ARG1 was similar. IAA5 transcription was twice as high at the bottom (# 5) than top (# 2) position. At 11 h, PIN1, PIN3, and ARG1 showed similar values for the convex and concave sites. The polarity of IAA5 was less, but still reached about five times the value of the top flank. After 48 h, the values were reversed and PIN1, PIN3 and ARG1, the polarity between top and bottom side was about six to eight-fold; the polarity for IAA5 was weaker. These data show that after reversal of curvature the transcription of the examined genes reached a comparable but reversed polarity as after 11 hours.
Recommended that the legend for Figure 7 specify that data corresponds for the average value of 3 probes on each hypocotyl flank. We have added the following sentence: The data for SPGE top and bottom refers to the average values of the combined probes 1, 2, 3 and 4, 5, 6, respectively.
Asked to include symbols indicating statistical significance for each graph. We have done so by including indicators for the least significant difference (LSD) on Figures 4, 5, 6, and 8. The remaining figures do not have statistically significant differences and therefore LSD was not included. We chose this approach because of the complexity of the data sets – individual symbols cannot adequately assess time, position, and genes.
Requested the variety of Brassica rapa. We included Brassica rapa v. rapa in line 339.
Reviewer 3 Report
The publication is well prepared, and I have only minor comments:
Line 3, 147, 153, 225, 226, 327, 372 Brassica rapa please change it to italics
I missed the main conclusion part in the publication.
Author Response
Reviewer 3
Reminded to use the italicized version of Brassica rapa We have italicized the name throughout the manuscript.
Mentioned that the main conclusion was missing. We have added the following text:
- Conclusion
Our results demonstrate that the basal hypocotyl curvature depends on auxin mediated process and involves the same genes for downward or upward curvature that were found to be involved in root curvature. The expression trend for auxin related genes in positively gravitropic hypocotyl curvature reverses when the hypocotyls begin to develop negative gravitropism. High resolution sampling revealed spatial variation in the transcription of auxin related genes throughout the curvature with greatest polarity present at the most concave and convex position in the curvature.